

# Variability of bulb morpho-geometrical properties in onion (*Allium cepa* L.) germplasm collections, using digital imaging tools

Seong-Hoon Kim[1,*], Kanivalan Iwar[1], JiWon Han[2], Inchan Choi[3], Jaesu Lee[4] and Kingsley Ochar[1,*]

[1] National Agrobiodiversity Center, National Institute of Agricultural Sciences, Rural Development Administration, Jeonju, Jeollabuk, Republic of Korea
[2] National Institute of Horticultural and Herbal Sciences, Rural Development Administration, Muan, Republic of Korea
[3] Department of Agricultural Engineering, National Institute of Agricultural Sciences, Rural Development Administration, Jeonju, Jeollabuk, Republic of Korea
[4] Korea Partnership for Innovation of Agriculture, RDA, Jeonju, Jeollabuk, Republic of Korea
[*] These authors contributed equally to this work.

Corresponding authors
Seong-Hoon Kim,
shkim0819@korea.kr
Kingsley Ochar, ocharking@korea.kr

## ABSTRACT

**Background**. Phenotypic characterization of onion germplasm is requisite for designing breeding programs, and for meeting industrial processing, and marketing demands. Onion bulb morphology, and geometrical properties, which are the physical and spatial dimensions and shape characteristics influence consumer and market demand, as well as suitability for processing and mechanizing post-harvest handling. Many previous studies employed manual tools such as Vernier calipers for measurement of onion bulb parameters, which is time-consuming. The emergence and application of phenomics tools such as digital cameras are more convenient for rapid phenotypic characterization.
**Aim**. This study aimed to investigate the phenotypic variability of 29 onion accessions based on ten qualitative and twelve quantitative bulb characteristics.
**Methodology**. Freshly harvested onion bulbs ($n = 10$/accession) were obtained from the Allium Vegetable Research Institute (AVRI), at Muan-Gun, Republic of Korea. A digital camera was used to capture images of the bulbs. The images were saved in JPEG file format, and uploaded into ImageJ software for measurement of linear dimensions, including polar diameter, equatorial diameter, transverse diameter or thickness. To ensure accurate measurement, images were first calibrated, using the straight line tool and the "Set scale" function in the software. Results of the linear dimensions were then used for estimating other geometrical properties, such as aspect ratio, sphericity, and geometric and arithmetic mean diameters.
**Results**. Our findings revealed a broad range of phenotypic variation within the germplasm. Polar and equatorial diameters ranged from 4.731 to 11.998 cm, and from 4.54 to 10.196 cm, with mean values of 9.213 and 7.472 cm, respectively. Also, geometric and arithmetic mean diameters ranged from 4.224 to 10.484 cm, and from 4.257 to 10.569 cm, with corresponding mean of 7.901 and 7.980 cm, respectively. Principal component analysis grouped the accessions into three distinct clusters, with cluster three composing the highest number of accessions. Strong significant positive associations were observed among several traits. For instance, polar diameter correlated
strongly with polar diameter and transverse diameter ($r > 0.97$), geometric and arithmetic mean diameters ($r > 0.98$), surface area ($r > 0.96$), frontal surface area ($r > 0.94$), cross sectional area ($r > 0.96$), equatorial diameter ($r > 0.83$), and thickness of neck ($r > 0.84$). High to moderate broad sense heritability and genetic gain were estimated for several traits.

**Conclusion**. Overall, the significant variability within the onion germplasm provides a potential for breeding new cultivars to meet consumer and industrial requirements. The results also provide information vital for future genomic and metabolite studies.

# INTRODUCTION

*Allium* is one of the most economically significant genera of the plant family, Amaryllidaceae, comprising >800 species (*Ochar & Kim, 2023*). These species include vegetables, such as onion (*Allium cepa* L.), green onion (*Allium fistulosum* L.), garlic (*Allium sativum* L.), chive (*Allium schoenoprasum* L.), leek (*Allium ampeloprasum* var *porrum*), and shallot (*Allium ascalonicum* L.) (*Kim et al., 2023c*; *Ochar & Kim, 2023*). Among *Allium* species, onion represents one of the earliest domesticated vegetables (*Mahajan & Gupta, 2023*; *Shankar et al., 2023*), and currently under extensive cultivation worldwide (*Elattar et al., 2024*; *Ochar & Kim, 2023*). In terms of global production and consumption, onion is ranked second most important vegetable, only surpassed by tomato (*Ndiaye et al., 2024*; *Sansan et al., 2024*), while among spicy vegetables, it is the most consumed globally. It is an important cash-generating vegetable crop, and contributes to economic prosperity of growers, domestic and international traders and industrial processors (*Yeshiwas, Alemayehu & Adgo, 2024*). In 2022, an estimated 110.62 million tons of onions were produced covering over 5.97 million hectares of land. India, China, Egypt, United States, Bangladesh, Turkiye, Pakistan, Indonesia, Iran, and Algeria are the leading producers of global dry bulb onions (*Sansan et al., 2024*). Onion bulbs form an integral component of many individual daily dietary intake across the globe (*Ndiaye et al., 2024*). They can be consumed as both fresh and cooked vegetables or processed into different products, such as dehydrated onion flakes, onion powder, pickles, and sauces (*Shankar et al., 2023*). Nutritionally, onion bulbs are rich in carbohydrates, dietary fiber, vitamins, and essential minerals like potassium, calcium, and magnesium (*Amir et al., 2023*; *Mahmood et al., 2021*; *Remi, 2023*). Onion bulbs also contain many bioactive compounds, such as flavonoids, polyphenols, and sulfur-containing compounds, which are essential based on their health-promoting benefits (*Sagar et al., 2022*; *Stoica et al., 2023*). As a result, onion bulbs are one of the most essential raw materials in functional food and pharmaceutical industries. Additionally, specific bioactive compounds in onion bulbs are used in agricultural bio-pesticides and cosmetic products (*Kumawat et al., 2014*; *Messias et al., 2023*; *Stoica et al., 2023*).

In food processing industries, bulb morphology and geometrical properties, which are the physical and spatial dimensions and shape characteristics, are important parameters
for determining the suitability of onion for processing (*Kaveri & Thirupathi, 2015*). These properties, encompassing bulb size, shape, color, and surface area are often assessed to evaluate onion quality, yield potential, and suitability for storage, packaging, and diverse uses, such as slicing or processing (*Kaveri & Thirupathi, 2015*). They also serve as important parameters in mechanizing post-harvest handling, as they can influence machine sorting, peeling, and cutting processes (*Kiran et al., 2024*). For instance, onion bulbs that are uniformly shaped and sized are preferred for mechanical planting and harvesting, as well as processing industries, since this facilitates reduced post-harvest losses and improved processing efficiency. Bulb size and shape influence storage capabilities as certain geometries are known to be more resistant to mechanical damage and sprouting. Elongated or oval-shaped bulbs are considered to be more preferred for slicing, while spherical bulbs are ideal for dicing or whole-bulb storage. Large-sized bulbs with firm outer skins tend to have better storage potential, as they are less prone to mechanical damage and sprouting. Additionally, small-sized bulbs are more commonly used for pickling due to their uniformity and ease of processing. Skin color, an important phenotypic trait, is also critical as it influences consumer preference (*Kiran et al., 2024*).

The genetic resources of onion comprise a broad diversity of germplasm, including landraces, wild relatives, and cultivated varieties, and represent a critical repository for breeding programs aimed at enhancing onion yield productivity, and quality, as well as for improving stress resilience of the crop (*Iwar et al., 2024*; *Ochar & Kim, 2023*). The phenotypic variability of onion bulb morphology and geometrical properties differ considerably within the genetic resources of the crop (*Kaveri & Thirupathi, 2015*). This variability presents opportunity for onion breeding to adapt cultivars to meet the specific needs of local and global markets (*Ochar & Kim, 2023*). By exploring the diversity present in genetic resources, breeders can select traits that improve not only agronomic performance but also those that can promote post-harvest quality. For instance, traits such as thinner necks and thicker outer skins are known to be better in extending shelf life and improve handling during transportation. Nonetheless, variability of bulb morpho-geometrical properties within diverse germplasm is not adequately characterized. In spite of previous studies on onion bulb morphological traits, limited research has focused on combined analysis of morphological and geometrical properties to guide selection and breeding for tailored consumer, market and industrial needs in regard to the crop's genetic resources. In this study, the exploration of phenotypic variability in onion germplasm bridges this gap, by providing a foundation for the development of improved cultivars that align with the demands of consumers and industry stakeholders. By integrating morphological and geometric trait analysis into breeding programs, onion production can be optimized to enhance efficiency, reduce waste, and meet the challenges of modern agriculture and food systems. Many previous studies employed manual tools, such as Vernier calipers for measurement of onion bulb parameters, especially the linear dimensions, which is time-consuming (*Gautam et al., 2023*; *Kumawat & Raheman, 2023*). High-throughput phenotyping technologies provide a better approach for rapid and more accurate measurements of plant traits (*Kim, Choi & Kim, 2024a*; *Kim, Subramanian & Hahn, 2023a*; *Kim et al., 2023b*). For instance, currently, there are advanced methods that

utilize imaging systems, such as 3D scanners and digital cameras, to capture detailed morphological data of plant organs in real time (*Kim & Kim, 2011*; *Kim et al., 2022*). These tools can provide a quicker approach to extract key phenotypic information, such as bulb size, shape, and volume, without the use of manual techniques. Therefore, the current study explored the phenotypic variability among 29 selected onion germplasm, with a focus on bulb characteristics, using digital cameras and the ImageJ software. This approach is more convenient for rapid phenotypic characterization of onion bulbs.

## MATERIALS & METHODS

### Onion Germplasm and description of experiment

Twenty-nine (29) onion accessions (Table S1) were used to investigate variability of bulb morphological properties. The germplasm was obtained from the Rural Development Administration (RDA) Genebank, located at the National Agrobiodiversity Center, National Institute of Agricultural Sciences, Jeonju, Republic of Korea. The accessions were selected from more than 500 germplasm resources cultivated at the Allium Vegetable Research Institute (AVRI), located at Muan-Gun (34°59′25.63 44″N, 126°28′54.0696″E, 31.48 m above sea level). Details of climate, soil characteristics, seedling development, planting, and cultivation practices adopted were similar to the previous study by *Luitel et al. (2023)*. The experimental area is characterized by a humid subtropical climate, with annual of precipitation of 1,000–1,800 mm. The soil type at the study area is clay, with a pH value of 5.5 to 7.2 (http://soil.rda.go.kr). The experiment was conducted from September, 2023 to June 2024. For the experimental laid out, the randomized complete block design (RCBD) with three replications was used. Cultivation practices according to the RDA recommendation for onions were followed. Seeds of each germplasm were first sown in early September, using plug trays, filled with a commercial growing medium (Plant world, Nongwoobio Co. Ltd., Suwon, Korea). Based on fertilizer application recommendation of the Rural Development Administration (*RDA, 2010*), compost (20 mg ha$^{-1}$) and pre-plant fertilizers, including 80 kg ha$^{-1}$ N (urea), 33.6 kg ha$^{-1}$ of $P_2O_5$ (fused phosphate), and 58 kg ha$^{-1}$$K_2O$ (potassium sulphate) were applied to the experimental plots. Pest control measures was followed, by applying terbufos and fosthiazate at the rate of 1.50 kg active ingredient per hectare (a.i./ha) to reduce damage resulting from maggots and nematodes. For planting beds, the center-to-center distance of 1.50 m with a 1.0 m bed width and 0.20 m height was used. On each bed, alchlor (43.7%) and pendimethalin (31.7%) were sprayed, before. Later, black plastic mulch was used on the bed, before transplanting of seedlings.

### Phenotypic measurement of bulb traits

Freshly harvested bulbs, selected from each accession ($n = 10$) were used to investigate the phenotypic variability of nine qualitative and 12 quantitative traits. All the qualitative and four quantitative (polar diameter, equatorial diameter, transverse diameter, and thickness of neck) traits were measured based on the guidelines of the descriptors of the International Union for the Protection of New Variety of Plants (*International Union for the Protection of New Varieties of Plants (UPOV), 2008*) (https://www.upov.int/edocs/tgdocs/en/tg046.pdf;

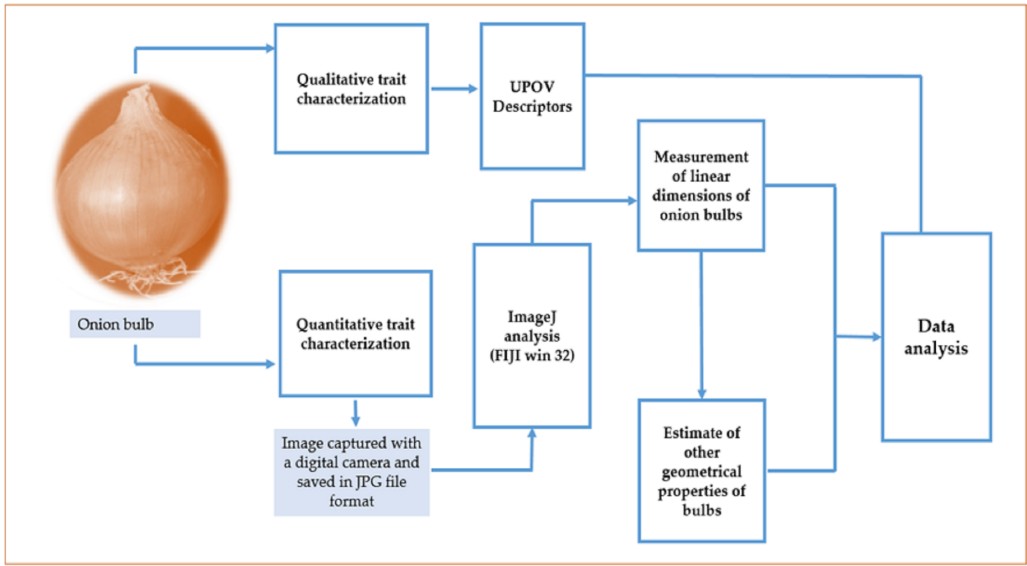

**Figure 1** **Workflow illustrating the procedure for the measurement and analysis of onion bulbs.**

Table S2). For the measurement of the quantitative traits, a digital camera was used as described in previous studies in soybean by *Kim et al. (2022)*. In brief, a macro lens (SEL30M35 E30 mm F3.5 Macro; Sony, Tokyo, Japan) was mounted on the body of the camera (α-6000; Sony), with the camera luminance set to 1/8 (F11 ISO 125). A CN-T96 light (ProDean; 832 lux) and VILTRONX light (VL-D85T; 2521 lux) were used to eliminate shadows from the bulbs. Through the crop and scale functions, only the area with bulbs in the 2D image was enlarged. The camera was positioned 15 cm directly above the bulbs at a 90-degree angle to achieve uniform imaging conditions. Noise was then removed, and hue saturation value (HSV) and binary values obtained in order to verify if the onion bulb and background were adequately separated. The images of onion bulbs captured were saved in JPEG file format, and uploaded into the ImageJ software, where the linear dimensions, including polar diameter, equatorial diameter, transverse diameter or thickness and thickness of neck were measured. To ensure accurate measurement, the images were first calibrated, using the straight line tool and the "Set scale" function in the software. Polar diameter was measured as the distance from the crown to the point of root attachment to the onion bulb, while equatorial diameter was measured as the maximum width of the onion in a plane perpendicular to the polar diameter (*Dabhi & Patel, 2017*), using the line selection tool in ImageJ. These linear dimensions formed the basis for estimating other geometrical parameters of the bulb (Table S3). Figure 1 illustrate the methodology used for measurement of the onion bulb parameters.

## Statistical analysis

The data used in the analysis were based on measurement of each geometric parameter of onion bulb in three replications. Analysis of variance (ANOVA) was carried out to examine the effects of accession on bulb parameters, using SPSS software (version 16; SPSS

Inc., Chicago, IL, USA). The degree of significance of each variable was tested using 5% probability. The variance component format was used to estimate variances, heritability and genetic advance based on the procedure used by *Amir et al. (2023)*. Descriptive statistics, frequency, and correlation analyses were performed with R software (version R-4.4.1). Descriptive statistics analysis (minimum, maximum mean, standard deviation, and coefficient of variation) was conducted to summarize the onion bulb parameters studied. Frequency analysis was carried out to assess the distribution of qualitative traits among the accessions, while correlation analysis was performed to determine the relationships between the quantitative traits. Multivariate analysis, including principal component analysis (PCA) and dendrogram were performed using the SIMCA software (V13.3, Umetrics, Sweden). The PCA was used to identify the key traits contributing to variation among accessions, while clustering analysis (HCA) was applied to generate a dendrogram for grouping accessions based on similarity.

## RESULTS

### Variability of qualitative traits of onion bulbs

In this study, a wide range of variability was observed among the qualitative traits of onion bulbs as shown in Table 1 and Fig. S1. Among the 29 accessions studied, bulb splitting tendency ranged from week to strong, with majority (69%) showing medium splitting tendency. Bulb size among the accessions was either small or medium, of which 25 (86%) accessions were identified as medium–sized. The position of maximum diameter of most of the accessions (20 accessions, 69%) was found in the middle of the bulb. Bulb neck width varied among the accessions, ranging from broad, medium, narrow, and very narrow. Thirteen accessions (45%) had broad bulb neck width. Bulb shape (longitudinal section) was categorized as broad elliptic, circular, elliptic, transverse medium elliptic, broad ovate, and rhombic. Broad elliptic and circular bulb shapes were more prevalent within the germplasm. Shape of stem end also varied considerably from round to flat, with 16 rounded (55%) and eight slightly raised (28%), composing majority of the accessions. For bulb shape at root end, accessions were mainly classified as rounded (19 accessions, 66%). The remaining ones were either weakly or strongly tapered. The base color of dry skin was categorized as red, brown, white, green, yellow, and pink, with brown (11 accessions, 38%) and red (nine accessions, 31%) colors dominating within the germplasm. Hue of color of dry skin were mainly brownish (11 accessions, 38%), reddish (eight accessions, 28%), absent (four accessions, 14%) and greenish (three accessions, 10%).

### Variability of quantitative traits of onion bulbs

The ANOVA (Table 2), descriptive statistics (Table 3) and frequency distribution analyses (Fig. S2) revealed a wide range of phenotypic variation among the 29 onion accessions. The coefficient of variation (CV) ranged from 7.670 to 32.794% for sphericity of bulb and frontal surface area, respectively. Polar and equatorial diameters of bulb differed among the accessions, ranging from 4.731 (accession 23P127) to 11.998 cm (accession 23C68) and from 4.354 (accession 23P127) to 10.196 cm (123C68), with mean values of 9.213 and 7.472 cm, respectively. Aspect ratio ranged from 0.921 (accession 230368) to 1.637

**Table 1  Range of variation of qualitative traits of onion bulb.**

| Trait | Code | Frequency | Relative frequency (%) |
|---|---|---|---|
| Tendency to split into bulblets | Medium | 20 | 69 |
| | Strong | 1 | 3 |
| | Very strong | 6 | 21 |
| | Weak | 2 | 7 |
| Degree of splitting into bulblets | Medium | 20 | 69 |
| | Strong | 1 | 3 |
| | Very strong | 6 | 21 |
| | Weak | 2 | 7 |
| Bulb size | Small | 4 | 14 |
| | Medium | 25 | 86 |
| Position of maximum diameter of bulb | Middle | 20 | 69 |
| | Towards stem end | 8 | 28 |
| | Towards root end | 1 | 3 |
| Bulb neck width | Medium | 8 | 28 |
| | Broad | 13 | 45 |
| | Narrow | 4 | 14 |
| | Very narrow | 4 | 14 |
| Bulb shape (longitudinal section) | Broad elliptic | 12 | 41 |
| | Circular | 12 | 41 |
| | Elliptic | 1 | 3 |
| | Transverse medium elliptic | 1 | 3 |
| | Broad ovate | 1 | 3 |
| | Rhombic | 2 | 7 |
| Shape of stem end | Rounded | 16 | 55 |
| | Slightly raised | 8 | 28 |
| | Slightly slopping | 2 | 7 |
| | Strongly slopping | 1 | 3 |
| | Flat | 2 | 7 |
| Shape at root end | Weakly tapered | 6 | 21 |
| | Rounded | 19 | 66 |
| | Strongly tapered | 4 | 14 |
| Base color of dry skin | Red | 9 | 31 |
| | Brown | 11 | 38 |
| | White | 4 | 14 |
| | Green | 2 | 7 |
| | Yellow | 2 | 7 |
| | Pink | 1 | 3 |

**Table 1** (*continued*)

| Trait | Code | Frequency | Relative frequency (%) |
|---|---|---|---|
| Hue of color of dry skin | Absent | 4 | 14 |
| | Reddish | 8 | 28 |
| | Brownish | 11 | 38 |
| | Greenish | 3 | 10 |
| | Yellowish | 1 | 3 |
| | Pinkish | 2 | 7 |

(accession 230157), with mean value of 1.637. The transverse diameter, also known as bulb thickness (T) is the dimension between the equatorial and polar diameter surfaces of the bulb. Transverse diameter varied from 3.686 (accession 23P127) to 9.515 (accession 23C68), while thickness of neck varied from 1.15 (accession No.200, 9-2) to 2.745 (accession 230310). The average values recorded for transverse diameter and thickness of neck were 7.255 cm and 2.078 cm respectively. The range of values recorded for shape index and sphericity of onion bulbs were 0.697 (accession 230157) to 1.258 (accession 230368) and 0.845 (accession 230368) to 1.247 (accession 230157), with mean values of 0.923 and 1.043, respectively. Geometric and arithmetic mean diameters are estimated from three linear dimensional measurements of the bulb: polar diameter, equatorial diameter, and transverse diameter or thickness (*Kaveri & Thirupathi, 2015*). Among the accessions, geometric and arithmetic mean diameters ranged from 4.224 (accession 23P127) to 10.484 cm (accession 23C68) and from 4.257 (accession 23P127) to 10.569 cm (accession 23C68), with mean values of 7.901 and 7.980 cm, respectively. The surface area was estimated from geometric mean diameter, while frontal surface area, and cross sectional area were estimated from two or three of the linear dimensions, equatorial diameter, polar diameter, and transverse diameter. The surface area of the onion bulbs ranged from 56.347 (accession 23P127) to 345.674 cm$^2$ (accession 23C68), with average of value of 202.469 cm$^2$. The frontal surface area and cross sectional area ranged from 16.222 (23P127) to 95.925 cm$^2$ (accession 23C68), and 14.306 (accession 23P127) to 87.837 cm$^2$ (accession 23C68), with mean values of 55.592 and 51.623 cm$^2$, respectively.

## Principal component analysis of selected traits of onion bulb

The unsupervised PCA was used to elucidate the clustering pattern of the accessions and for identifying the variables that are most relevant or strongly correlated with each component (Table 4). The PCA was performed using twelve quantitative traits of onion bulbs. The first three principal components accounted for 70.7%, 26.0%, and 2.5% of the total variation, with standard deviation of 2.913, 1.767, and 0.549, respectively. Most of the traits contributed positively to principal component 1 (PC1). Traits, aspect ratio (−0.549) and sphericity (−0.557) showed significant negative contribution to principal component 2 (PC2), whereas all the other characters had positive contributions, with the highest contribution observed for shape index (0.546). For principal component 3 (PC3), the highest significant positive contribution was associated with thickness of neck

**Table 2   Statistical analysis of variance for quantitative traits of onion bulbs.**

| Variables | Groups | Sum of squares | df | Mean square | *F* value | Sig. |
|---|---|---|---|---|---|---|
| Polar diameter | Between groups | 61.641* | 11 | 5.604 | 2.110 | 0.039 |
| | Within groups | 119.489 | 45 | 2.655 | | |
| | Total | 181.131 | 56 | | | |
| Equitorial diameter | Between groups | 24.592NS | 11 | 2.236 | 1.047 | 0.424 |
| | Within groups | 96.115 | 45 | 2.136 | | |
| | Total | 120.707 | 56 | | | |
| Aspect ratio | Between groups | 0.961*** | 11 | .087 | 3.499 | 0.001 |
| | Within groups | 1.123 | 45 | .025 | | |
| | Total | 2.083 | 56 | | | |
| Transverse diameter | Between groups | 35.227* | 11 | 3.202 | 2.066 | 0.044 |
| | Within groups | 69.746 | 45 | 1.550 | | |
| | Total | 104.973 | 56 | | | |
| Thickness of neck | Between groups | 4.270*** | 11 | .388 | 3.835 | 0.001 |
| | Within groups | 4.556 | 45 | .101 | | |
| | Total | 8.826 | 56 | | | |
| Shape index | Between groups | 0.588*** | 11 | .053 | 6.745 | 0.000 |
| | Within groups | 0.357 | 45 | .008 | | |
| | Total | 0.945 | 56 | | | |
| Sphericity | Between groups | 0.301*** | 11 | .027 | 4.442 | 0.000 |
| | Within groups | 0.278 | 45 | .006 | | |
| | Total | 0.579 | 56 | | | |
| Geometric mean diameter | Between groups | 33.671NS | 11 | 3.061 | 1.640 | 0.120 |
| | Within groups | 83.978 | 45 | 1.866 | | |
| | Total | 117.649 | 56 | | | |
| Arithmetric mean diameter | Between groups | 34.072NS | 11 | 3.097 | 1.640 | 0.120 |
| | Within groups | 84.993 | 45 | 1.889 | | |
| | Total | 119.065 | 56 | | | |
| Surface area | Between groups | 77,400.249NS | 11 | 7,036.386 | 1.753 | 0.092 |
| | Within groups | 180,654.522 | 45 | 4,014.545 | | |
| | Total | 258,054.771 | 56 | | | |
| Frontal surface area | Between groups | 6,070.981NS | 11 | 551.907 | 1.750 | 0.093 |
| | Within groups | 14,189.657 | 45 | 315.326 | | |
| | Total | 20,260.638 | 56 | | | |
| Cross sectional area | Between groups | 4,984.208NS | 11 | 453.110 | 1.732 | 0.097 |
| | Within groups | 11,771.025 | 45 | 261.578 | | |
| | Total | 16,755.233 | 56 | | | |

**Notes.**
Level of significance: * $p < 0.05$; ** $p < 0.01$; *** $p < 0.001$; NS, Not significant.

(0.937). To understand the main quantitative traits contributing to variability in the onion germplasm, loading plots were used to compare traits in PC1 and PC2 (Fig. S3). The results showed that almost all of the quantitative traits (except shape index) showed a positive weight on PC1, which explained the largest proportion (70.7%) of the total variation.

**Table 3  Variation of quantitative characteristics of onion bulbs.**

| Quantitative trait | Designation | Minimum | Maximum | Mean | SD | CV |
|---|---|---|---|---|---|---|
| Polar diameter (cm) | PD | 4.731 | 11.998 | 9.213 | 1.693 | 18.376 |
| Equatorial diameter (cm) | ED | 4.354 | 10.196 | 7.472 | 1.395 | 18.670 |
| Aspect ratio | Ra | 0.921 | 1.637 | 1.242 | 0.143 | 8.735 |
| Transverse diameter (cm) | T | 3.686 | 9.515 | 7.255 | 1.340 | 18.470 |
| Thickness of neck (cm) | TN | 1.150 | 2.745 | 2.078 | 0.379 | 18.239 |
| Shape index | SHI | 0.697 | 1.258 | 0.923 | 0.111 | 12.026 |
| Sphericity | STY | 0.845 | 1.247 | 1.043 | 0.080 | 7.670 |
| Geometric mean diameter (cm) | Dgm | 4.224 | 10.484 | 7.901 | 1.401 | 17.732 |
| Arithmetic mean diameter (cm) | Dam | 4.257 | 10.569 | 7.980 | 1.413 | 17.707 |
| Surface area (cm$^2$) | Sa | 56.347 | 345.674 | 202.469 | 65.687 | 32.443 |
| Frontal surface area (cm$^2$) | Af.s | 16.222 | 95.925 | 55.592 | 18.231 | 32.794 |
| Cross sectional area (cm$^2$) | Ac.s | 14.306 | 87.837 | 51.623 | 16.688 | 32.327 |

Further, to investigate the relationships between quantitative and qualitative characteristics of the bulbs, and to elucidate those characteristics that contributed mainly to the total variation within the germplasm, PC1 and PC2 were compared, using loading plots (Fig. S3). The results revealed a number of bulb characteristics, exhibiting higher weight on PC1, including bulbs with maximum diameter position in the middle, bulbs with medium tendency and degree of splitting into bulblets, bulbs with shape that are strongly tapered at root end, bulbs with medium and very narrow neck width, bulbs with shape that are rounded and slightly slopping at stem end, and bulbs with brownish, reddish or no (absent) hue of color of dry skin. The contribution of bulb size, shape, and color of dry skin categories to the variability among the 29 onion accessions were also investigated, using PCA (Fig. 2). For bulb size categories (small and medium), PC1 explained significant proportion of the variability and was positively associated with nearly all accessions with medium sized bulbs (Fig. 2A). In contrast, PC2 was largely associated with accessions having small bulb size. Three accessions with medium-sized bulbs negatively correlated with PC2. Additionally, four accessions with medium size bulbs positively associated with PC2. The PCA of bulb shapes among the 29 onion accessions revealed a clear pattern of variability based on shape categories (Fig. 2B). Accessions with broad elliptic and circular bulb shapes, which are the dominant shapes within the germplasm, showed positive weights on PC1. Interestingly, accession 9, with a less dominant broad ovate shaped-bulb also aligned positively with PC1. Rhombic-shaped bulb, though associated with only two accessions, showed their weight on PC2. The contribution of dry bulb skin color categories to phenotypic variability within the onion germplasm was also elucidated based on PCA (Fig. 2C). Accessions with bulbs showing dominant colors of dry skin such as brown, red and white were clustered on the positive side of PC1. The association of green-colored dry skin bulbs with PC1, though limited to only two accessions further support their alignment with dominant color traits. Accessions with less dominant colors, such as pink and yellow were mainly linked to PC2.

PCA score plot is used to elucidate how different accessions are related based on certain characteristics (*Rodionova, Kucheryavskiy & Pomerantsev, 2021*). In this study,

**Table 4  Principal component analysis showing trait contribution to the first three PCs in onion germplasm.**

| Trait | Principal component (Eigenvectors) | | |
|---|---|---|---|
| | PC1 | PC2 | PC3 |
| Polar diameter | **0.338** | 0.077 | −0.002 |
| Equatorial diameter | **0.308** | 0.244 | −0.120 |
| Aspect ratio | 0.068 | **−0.549** | 0.090 |
| Transverse diameter | **0.335** | −0.086 | −0.201 |
| Thickness of neck | 0.294 | 0.013 | **0.937** |
| Shape index | −0.080 | **0.546** | 0.071 |
| Sphericity | 0.058 | **−0.557** | −0.060 |
| Geometric mean diameter | **0.342** | 0.028 | −0.113 |
| Arithmetic mean diameter | **0.342** | 0.022 | −0.104 |
| Surface area | **0.340** | 0.053 | −0.112 |
| Frontal surface area | **0.335** | 0.112 | −0.062 |
| Cross sectional area | **0.341** | 0.047 | −0.102 |
| Standard deviation | 2.913 | 1.767 | 0.549 |
| Proportion (%) | 70.7 | 26.0 | 2.5 |
| Cumulative (%) | 70.7 | 96.7 | 99.2 |

**Notes.**
The bold values indicate the highest loadings in the principal component analysis, and mainly accounted for the variability in the germplasm.

the PCA analysis of onion bulb traits among the 29 onion accessions provided insight into the contribution of variables and accessions to phenotypic variability (Fig. 3A). Cluster I, comprised only two accessions (accessions 23P127 and No. 200,9-2), with positive associated with PC2. These two accessions had broad elliptic bulb shape, and round shape of stem and root ends (Table 1). They also exhibited medium tendency and degree to split into bulblets, medium bulb size, and the position of their maximum diameters positioned at the middle. Cluster II, consisting of six accessions showed mixed associations, with four aligning to the positive side of PC1 and the remaining two accessions found on the negative side of PC2. Cluster III, which contained the majority of the accessions was associated with PC1.

In line with the PCA results, the dendrogram revealed three distinct clusters of accessions (Fig. 3B). Cluster I contained the same two accessions (accessions 23P127 and No.200, 9-2) as in the PCA, while Cluster II comprised six accessions, all in line with the PCA results. Cluster III was further classified into two sub-clusters, consisting of nine and 12 accessions each. Characteristically, all accessions in cluster I exhibited weak bulb splitting, pinkish hue color of dry skin, rounded shape at root end and small bulb size. They also exhibited close values for bulb sphericity, shape index and aspect ratio. Other observed characteristics among members in this cluster were thickness of neck (<2.0 cm), transverse diameter (<5.0 cm), equatorial diameter (<6 cm), polar diameter (<6 cm), geometric and arithmetic mean diameters (<6 cm) and transverse diameter (<5 cm). Accessions in Cluster II were characterized by very strong bulb splitting attribute, mostly reddish in terms of dry skin color hue, mostly rounded bulb shape at root end, mostly broad neck

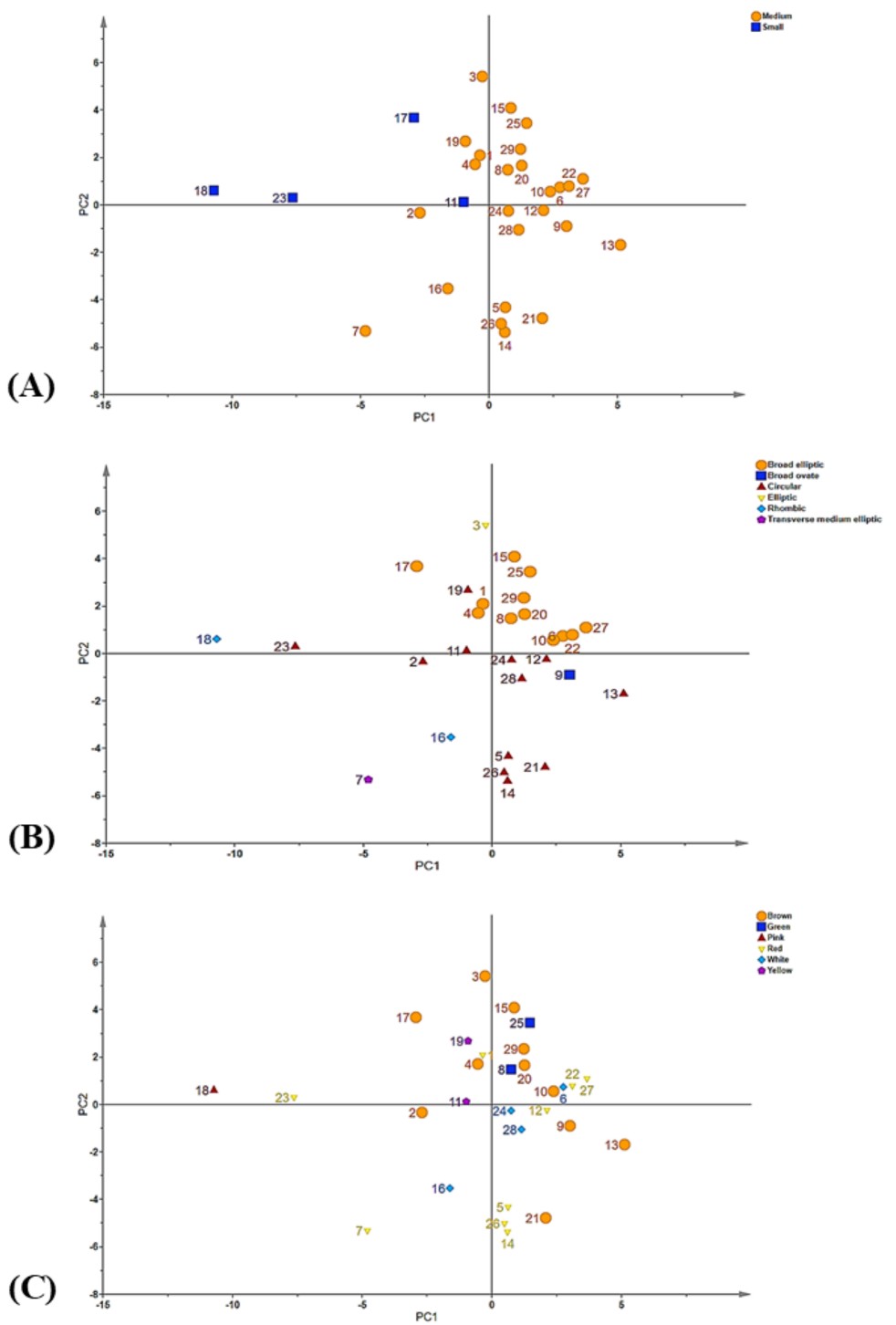

**Figure 2** Score plots of onion accessions based on size (A), shape (B) and color of dry skin (C) categories of the bulb.

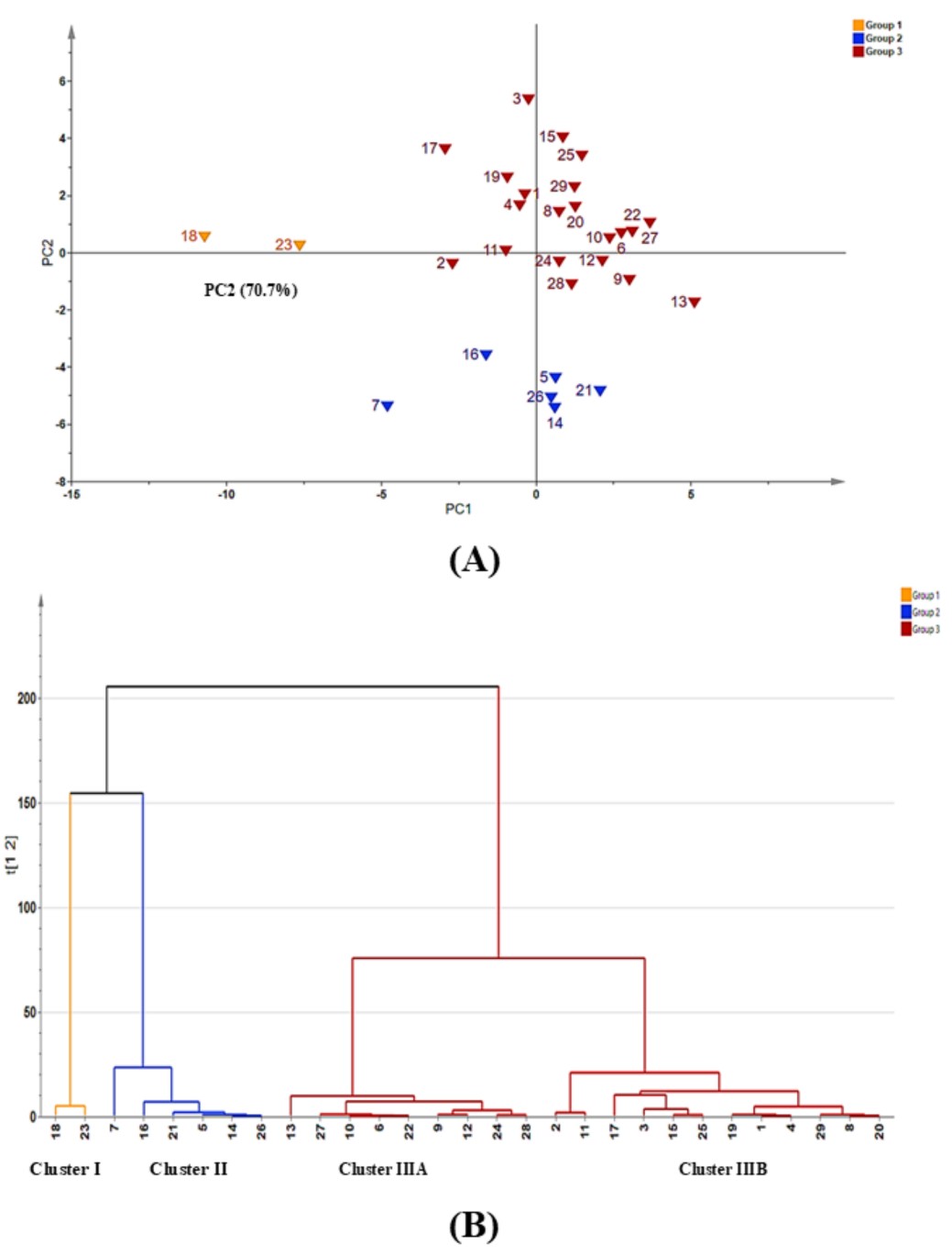

**Figure 3** PCA score plot (A), and dendrogram (B) showing clustering pattern of 29 onion accession.

width, medium bulb size, with position of maximum diameter located towards stem end. For the quantitative traits, most accessions in this cluster shared common traits such as equatorial and polar diameters (>8.0 cm), Geometric and arithmetic mean diameters (>6.0 cm), thickness of neck (>2.0 or near 2.0 cm), and transverse diameter (>7.0 cm). Bulbs of accessions in Cluster IIIA were medium sized and exhibited medium splitting
property. They also showed a combination of qualitative attributes for traits such as hue of color of dry skin (brown, reddish or absent), base color of dry skin (broad, red or white), neck width (broad, narrow or medium), and position of maximum diameter (mostly middle). For the quantitative traits, observed common characteristics included arithmetic and geometric mean diameters (>8 cm), bulb sphericity (>1.0), shape index (near 1.0), thickness of neck (>2 cm), transverse diameter (mostly >8 cm), equatorial diameter (>8.0 cm) and polar diameter (>10 cm). Accessions in Cluster IIIB exhibited varied forms of qualitative characteristics such as bulb splitting (medium, strong and very strong), dry skin coloration (mostly brown), bulb neck width (mostly medium and narrow), position of maximum diameter (mostly middle or toward stem end) and bulb size (medium or small). Common quantitative traits included arithmetic and geometric mean diameters (>7 cm), bulb sphericity and shape index (1 or near 1), thickness of neck (2 or near 2), transverse diameter (mostly > 6 cm), equatorial diameter (mostly > 6 cm) and polar diameter (mostly > 8 cm). Generally, shape- and area-related properties differed significantly within each cluster.

## Correlation analysis of morphological characters

The Pearson correlation coefficient was used to examine the association between the quantitative traits, with significant positive correlations observed for a number of pairs of traits (Fig. 4). For instance, polar diameter showed high positive correlation with transverse diameter ($r > 0.97$), geometric arithmetic mean diameters ($r > 0.98$), surface area ($r > 0.96$), frontal surface area ($r > 0.94$), and cross sectional area ($r > 0.96$). Equatorial diameter showed high positive correlation with frontal surface area ($r > 0.96$), surface area ($r > 0.93$), cross sectional area ($r > 0.92$), and geometric and arithmetic mean diameters ($r > 0.92$). High significant positive correlation was also observed between transverse diameter and geometric diameter ($r > 0.97$), arithmetic diameter ($r > 0.97$), surface area ($r > 0.96$), cross sectional area ($r > 0.96$), and frontal surface area ($r > 0.92$). Bulb shape index showed a strong negative significant correlation with aspect ratio ($r > 0.96$) and sphericity ($r > 0.99$). Also, very weak correlation was observed between aspect ratio and frontal surface area, thickness of neck, geometric diameter, arithmetic diameter, surface area, frontal surface area and cross sectional area. Generally, geometric mean diameter, arithmetic mean diameter, surface area, frontal surface area and cross sectional area exhibited strong and significant positive correlation among each other.

## Variance components and heritability

The variance components analysis revealed a wide range of variations among most of the onion bulb traits studied (Table 5). The highest genotypic coefficient of variation (GCV) and phenotypic coefficient of variation (PCV) were associated with surface area, frontal surface area, and cross section area. The highest heritability vales (>60%) were recorded for aspect ratio (71.26%), thickness of neck (73.97%), shape index (84.91%), and bulb sphericity (77.78%). The maximum genetic gain was estimated for thickness of neck (26.37%), shape index (25.40%), frontal surface area (21.64%), surface area (21.16%), cross sectional area (20.73%), and aspect ratio (20.13%).

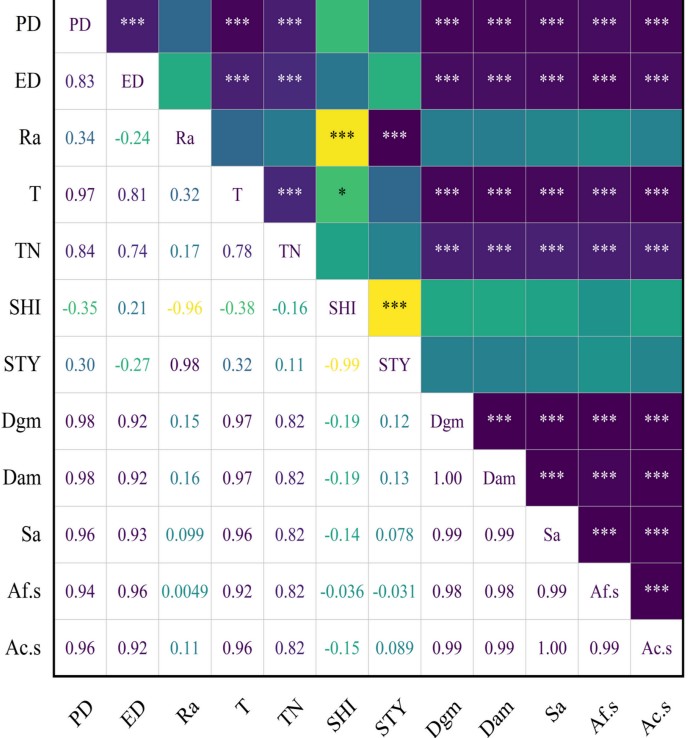
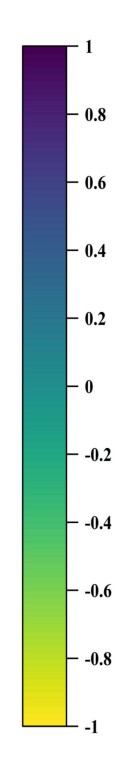

* p<=0.05  ** p<=0.01  *** p<=0.001

**Figure 4** **Pearson correlation among the quantitative traits of onion bulbs.** PD, Polar diameter; ED, Equatorial diameter; Ra, Aspect ratio; T, Transverse diameter; TN, Thickness of neck; SHI, Shape index; Sty, Sphericity; Dgm, Geometric mean diameter; Dam, Arithmetic mean diameter; Sa, Surface area; Af.s, Frontal surface area; Ac.s, Cross sectional area.

# DISCUSSION

## Phenotypic variability of qualitative traits of onion bulbs

Crop germplasm represent a valuable source of genetic diversity for investigating genetic and phenotypic variation, which is requisite for designing breeding programs (*Nguyen & Norton, 2020*; *Rivera et al., 2016*). Qualitative parameters of onion bulbs are important consideration in the marketability, storage and breeding of *Alliums* (*Gateri et al., 2018*). In this study, a wide range of variation was observed for the qualitative traits, reflecting a rich genetic diversity within the onion germplasm. For instance, bulb tendency and degree of splitting ranged from weak to strong, with medium splitting being the most prevailing within the germplasm. Bulb splitting tendency is a significant trait as it directly influences the marketability and consumer preference for onions. Onion bulb splitting can compromise the quality and size of the bulb, by reducing their commercial value and suitability for certain culinary uses. As a result, knowledge about this trait is essential in breeding programs aimed at improving bulb uniformity, storage quality, and adaptability to different growing conditions. Bulb size is an important breeding goal in onion

**Table 5  Estimate of variance components, heritability (h²), and genetic advance (GA) for quantitative traits of onion bulbs.**

| Trait | Grand mean | GV | PV | PCV (%) | GCV (%) | GCV/ PCV | h² (%) | GAM (%) |
|---|---|---|---|---|---|---|---|---|
| Polar diameter | 9.21 | 0.98 | 1.87 | 45.03 | 32.66 | 0.73 | 52.62 | 16.08 |
| Equitorial diameter | 7.52 | 0.03 | 0.75 | 31.49 | 6.66 | 0.21 | 4.47 | 1.06 |
| Aspect ratio | 1.24 | 0.02 | 0.03 | 15.28 | 12.90 | 0.84 | 71.26 | 20.13 |
| Transverse diameter | 7.24 | 0.55 | 1.07 | 38.39 | 27.58 | 0.72 | 51.59 | 15.16 |
| Thickness of neck | 2.08 | 0.10 | 0.13 | 24.95 | 21.46 | 0.86 | 73.97 | 26.37 |
| Shape index | 0.92 | 0.02 | 0.02 | 13.89 | 12.80 | 0.92 | 84.91 | 25.40 |
| Sphericity | 1.04 | 0.01 | 0.01 | 9.29 | 8.19 | 0.88 | 77.78 | 14.57 |
| Geometric mean diameter | 7.90 | 0.40 | 1.02 | 35.94 | 22.45 | 0.62 | 39.04 | 10.28 |
| Arithmetic mean diameter | 7.98 | 0.40 | 1.03 | 35.97 | 22.46 | 0.62 | 39.01 | 10.23 |
| Surface area | 202.47 | 1,007.28 | 2,345.46 | 340.36 | 223.05 | 0.66 | 42.95 | 21.16 |
| Frontal surface area | 55.59 | 78.86 | 183.97 | 181.91 | 119.10 | 0.65 | 42.87 | 21.54 |
| Cross sectional area | 51.62 | 63.84 | 151.04 | 171.05 | 111.21 | 0.65 | 42.27 | 20.73 |

(*Mahajan & Gupta, 2023*). Size of onion bulbs influences yield, quality and marketability of onion bulbs, with large-sized bulbs generally producing higher yield relative to smaller sizes (*Kamala et al., 2014*; *Mollah et al., 2015*). Therefore, in onion cultivation and marketing, production of onions with desirable bulb sizes is of a priority (*Ikeda et al., 2019*). Bulb size predominantly fell into the medium category (86% accessions). In onion, the maximum position of the diameter is the widest horizontal cross-section of the bulb, which varies depending on the bulb shape. Bulbs with the maximum position at the middle are generally spherical or globular, suggesting a uniform distribution of growth along the vertical axis. Such bulbs are usually preferred for their ease of handling, visual appeal, and consistent size, and are desirable for market and culinary applications. Knowledge about the position of maximum diameter is vital for breeders aiming to develop onion varieties with specific shapes to meet consumer preferences and market demands. Within the germplasm studied, for most accessions, maximum diameter was located at the middle of the bulb (69%). Neck width varied significantly, ranging from very narrow to broad, with broad necks observed in 45% of the accessions, which may influence storage and marketability. Like size, onion bulb shape influences consumer preference and suitability for processing (*Havey, 2024*; *Kamala et al., 2014*). Larger round bulbs are desirable in some markets and by some processing forms such as onion rings and diced products, while other consumers may prefer smaller and/or flatter bulbs (*Havey, 2024*). Bulb shape in terms of its longitudinal section was categorized into six forms, with broad elliptic and circular shapes dominating within the germplasm, and emphasizes the dominant morphological patterns among the accessions. Similarly, a previous study on phenotypic diversity and genetic variation in 23 diverse onion germplasm indicated that bulb shape exhibited different shape categories, ranging from flat to elliptic shape (*Neelam Sunil et al., 2014*). In this study, stem end shape also varied widely, with rounded (55%) and slightly raised (28%) forming the most frequently occurring characteristic. Bulb shape at the root end was mostly rounded (66%), with a smaller proportion being weakly or strongly tapered. Dry skin base colors demonstrated

notable variation, with brown and red as the most dominant, followed by white, yellow, pink, and green. The hue of the dry skin color was primarily brownish (38%) and reddish (28%), with only a few accessions exhibiting greenish or absent hues.

These results revealed significant phenotypic diversity within the onion germplasm, and provide valuable information requisite for future breeding programs aimed at improving bulb traits, such as improving onion marketability and storage life, as well as for different end uses. Bulb shape, such as spherical and globular forms may be prioritized in breeding for enhanced bulb uniformity, ease of handling, and visual appeal, thus making them more attractive to consumers and markets. For instance, in dehydration industries, onion bulb with a characteristic globose shape, white colored, and thin neck are usually preferred (*Mahajan et al., 2021*). Similarly, bulb splitting tendency could be minimized *via* selection, since excessive splitting decreases post-harvest quality and storage longevity. In terms of size, selecting for larger bulb sizes with a balanced maximum position of the diameter can improve processing efficiency and consumer preference. Additionally, breeding efforts can focus on maintaining broad elliptic and circular bulb shapes, which were predominant in the studied germplasm, to ensure consistency in storage and culinary applications while improving adaptability to market demands. Qualitative traits of onion bulbs are known to be correlated with variation of chemical composition, including total soluble solids (*Major et al., 2023*), thus the current germplasm provide enormous potential for biochemical investigation in onion. By selecting traits such as splitting tendency, shape, and bulb color, both market appeal and storage life can be enhanced. For instance, darker-colored bulbs are often associated with higher antioxidant content, making them more desirable for health-conscious consumers, while spherical bulbs with minimal splitting improve uniformity, handling, and post-harvest longevity. By breeding for these traits, onion varieties can be developed to maximize both consumer preference and biochemical stability, which can further ensure better storage quality and market competitiveness.

## Phenotypic variations of quantitative traits of onion bulbs
### Linear dimensions of onion bulb

The linear dimensions of onion bulbs, comprising polar diameter, equatorial diameter, neck diameter, and bulb thickness or transverse diameter are important properties of onions as they influence marketability, storability, processing efficiency, and consumer preferences. They are essential in determining suitability and quality of onion for various purposes, including mechanical grading, packaging, and processing (*Gautam et al., 2023*). These traits form the basis for estimating several geometrical parameters such as aspect ratio, bulb shape index, bulb sphericity, geometric mean diameter, arithmetic mean diameter, and bulb surfaces. Generally, linear dimensions of onion bulbs vary significantly across different genotypes. In the present study, polar diameter, equatorial diameter and transverse diameter ranged from 4.731 to 11.998 cm, 4.354 to 10.196 cm, and 3.686 to 9.515 cm, respectively, which indicate enormous variation within the selected germplasm. A polar and equatorial diameter varying from 5.18–11.96 cm and 5.87–11.08 cm was recorded in a previous study (*Singh et al., 2020*). In a study that investigated the genetic variability and diversity of selected red onion germplasm, a wide range of variation among

the onion genotypes was recorded, with polar diameter ranging from 2.31 to 3.08 cm and mean of 2.66 cm (*Amir et al., 2023*). In another study, involving genetic variability and diversity in onion, values for polar and equatorial diameters, ranged from 4.28–5.98 cm and 4.30–6.61 cm, respectively (*G, RP & Patil, 2021*). Different values for polar and equatorial diameters were also recorded in others studies that investigated engineering properties of selected onions for designing onion harvester (*Kumawat & Raheman, 2023*). The observed variation in bulb dimensions have important implications for crop breeding and market selection, and capable of influencing production efficiency and commercial value. Variability in polar and equatorial diameters provides opportunities to develop onion varieties suitable to specific market segments, with larger bulbs usually favored in fresh markets and medium-sized bulbs preferred for processing and dehydration. The wider range of bulb sizes also affects uniformity, which is a key trait for mechanized harvesting, packaging, and consumer preference. Neck thickness, an important trait for storage and post-harvest quality (*Singh et al., 2020*), especially in bulb curing process and disease prevention during storage, making this trait a crucial target in breeding (*Kamala et al., 2014*). In this study, this trait varied from 1.150 to 2.745 cm within the germplasm, compared to results of previous results where bulb neck thickness ranged from 0.78–1.38 cm (*G, RP & Patil, 2021*) and 0.42–2.46 (*Singh et al., 2020*). By selecting for optimal bulb size and neck thickness, breeding programs can be designed to target developing onion varieties that are high-yielding, market-oriented, and that which meet industry demands for storage stability, processing suitability, and consumer appeal.

### Shape-related properties of onion bulbs
Aspect ratio or shape factor, bulb shape index (the ratio of polar to equatorial diameters), and sphericity are important geometrical properties that influence market preference and storage properties. Aspect ratio, the ratio of the polar diameter to the equatorial diameter is often used as indicator for assessing bulb shape uniformity, and helps classify bulbs according to their shapes (*Kumawat & Raheman, 2023*). Results of the current study revealed considerable variation of aspect ratio within the germplasm. Similarly, shape index and sphericity values, which influence bulb appearance, also exhibited substantial variation among the accessions, varying from 0.697 to 1.258 and 0.845 to 1.247. As onion bulbs with shape index <1.5 are often considered as spherical in shape (*Kaveri & Thirupathi, 2015*), the accessions used in this study were mainly spherical. In a previous study, values for shape index among 53 diverse germplasm, ranged from 0.82–1.16 cm (*G, RP & Patil, 2021*). Sphericity is a measure of how closely the shape of an onion bulb approaches that of a perfect sphere, with bulbs having a sphericity of 1.0 usually considered as spherical (*Kumawat & Raheman, 2023*). In this study, sphericity value ranged from 0.0845 to 1.247 cm, with a mean value of 1.043. Closely related values were previously reported, with mean values of 1.2 (*Gautam et al., 2023*). The presence of variability of these traits among the accessions is important for future breading and genomic studies.

### The geometric and arithmetic mean diameters
Geometric and arithmetic mean diameters, estimated from the linear dimensions of the bulb are crucial for assessing the size, shape, and uniformity of bulbs (*Dabhi & Patel, 2017*).

They are ideal for evaluation and comparison of bulb dimensions across different varieties, and are crucial for designing breeding programs, grading standards, optimizing storage and packaging, and marketability of onion bulbs (*Khura, Mani & Srivastava, 2010*; *G, RP & Patil, 2021*). In engineering studies, geometric and arithmetic mean diameters are used for determining spacing between separator rods of onion separator (*Gautam et al., 2023*). The results of this study indicated that the geometric and arithmetic mean diameters exhibited closely related ranges, from 4.224 to 10.484 cm and from 4.257 to 10.569 cm, respectively, demonstrating a strong correlation in the distribution of bulb sizes.

### Bulb surface-related traits

Bulb surface area, especially, frontal, and cross-sectional areas are crucial for evaluating bulb development, estimating marketable yield, and designing efficient post-harvest handling and processing systems. Frontal, and cross-sectional areas represent the visible area when the bulb is viewed from the top or bottom, and the area exposed when the bulb is cut perpendicular to its polar diameter, respectively (*Gautam et al., 2023*). These traits are important for assessing quality, uniformity, visual grading and packaging of onion bulbs. Also, their variation in a set of germplasm allows breeders and agronomists to develop and cultivate onion varieties tailored to specific market demands and processing requirements. A broad range of variability was observed for these traits across the accessions studied. In this study, surface area ranged from 56.347 to 345.674 $cm^2$, with mean value of 202.469 $cm^2$. The range of values for frontal and cross section areas were 16.222 to 95.925 $cm^2$ and 14.306–87.837 $cm^2$ respectively, while their corresponding average values were 55.592 and 51.623 $cm^2$. In a related study, mean values for bulb surface area, frontal surface area and cross sectional area were 1,965.96, 7,814.06, and 1,649.11 $cm^2$, respectively (*Gautam et al., 2023*). The differences in values may be the result of different accessions used, and environmental factors.

## Multivariate analysis of bulb morphological traits

PCA provides an important means for elucidating relationships and variations within diverse germplasm (*Kim et al., 2024b*; *Singh et al., 2013*; *Sudha et al., 2019*). It also helps in identifying the most significant characters associated with individual accessions, and pinpointing traits that are critical contributors to observed variation within specific germplasm (*Bal, Maity & Sharangi, 2021*; *Dangi et al., 2018*). Based on the PCA, the major sources of variability within the onion germplasm were visualized. The two highest-ranking PCs, PC1 and PC2 explained 70.7% and 26% of the variation respectively, collectively accounted for 96.7% of the total variance. The PCA revealed significant phenotypic variability in the onion germplasm, with most quantitative traits (except shape index) contributing loading heavily on PC1. Loading plots of PC1 and PC2 provided valuable insight into bulb morphological traits that drive variation within the germplasm. Traits such as the position of maximum diameter (middle), medium tendency and degree of splitting into bulblets, strongly tapered root ends, medium and very narrow neck widths, rounded and slightly sloping stem ends, and brownish, reddish, or absent dry skin hues were major contributors to PC1. This implies that these traits are important for distinguishing

different onion genotypes within the germplasm. The contribution of dry skin hue to PC1 suggests its importance in germplasm differentiation, particularly in markets where onion color influences consumer preference and commercial grading. Bulb size categories (small and medium) played a pivotal role in driving variability. Previous studies have shown that, traits having the largest absolute value nearer to unity within the PC1, mainly influence the clustering relative to those with lower absolute values that are closer to zero (*Singh et al., 2013*). PC1 showed a strong positive correlation with genotypes characterized by medium-sized bulbs, suggesting that this trait as key driver of variability along this component. The strong positive correlation of PC1 with medium-sized bulbs implies that this size category plays a dominant role in structuring the variation, which may be relevant for breeding programs aimed at developing varieties suitable for specific market preferences. In contrast, PC2 primarily explained variability in small bulb sizes, which underscores their unique contributions to the observed variation within the germplasm, and thus plays a distinct role in differentiating accessions within the germplasm. Interestingly, three medium-sized and four medium-sized bulb accessions associated with PC2, which suggest a partial overlap in traits contributing to variation along both components. This partial overlap of some medium-sized bulb accessions with PC2 reveals certain morphological traits, such as neck thickness or bulb shape, may contribute to variability along both PC1 and PC2, supporting the complexity of trait interactions. The association of accessions of small bulb size with PC2 indicates the distinct contribution of bulb size to variability within the germplasm. These accessions possess unique attributes which contribute to their differentiation. This observation suggests that medium-sized bulbs contributed more broadly to phenotypic diversity, while small bulb size accounted for variability in more targeted manner. The results highlight the potential of bulb size as a discriminative trait in categorizing and elucidating the phenotypic variability among the studied varieties, and particularly valuable for breeding programs targeting size-specific onion varieties for different market segments. The PCA of bulb shapes showed a strong pattern of variation based on classifications. Broad elliptic and circular bulb shapes were the most prevalent within the germplasm and strongly aligned with PC1, suggesting that these traits play a major role in structuring variation within the germplasm. These forms are dominant traits and could be prioritized in breeding programs aimed at improving uniformity and market preference. Less prevalent shapes, such as broad ovate and rhombic had positive load on PC1. This observation indicates a shared phenotypic trait with dominant shapes and underscores the importance of rare shapes in maintaining genetic diversity. Dry bulb skin color categories also played a key role in phenotypic variation. Dominant colors like brown, red, and white showed a positive load on PC1, highlighting their importance in influencing primary variation, while rare colors exhibited dual associations with both PC1 and PC2. The dual association of dominant and rare colors traits with PC1 and PC2 emphasize the genetic diversity within the germplasm, which could be exploited in developing new onions cultivars with enhanced consumer appeal. Thus, color of dry bulb skin contributes significantly to the phenotypic variation of onion germplasm, while the less dominant colors are unique traits that enhance genetic diversity. The PCA score plots revealed the clustering pattern of the genotypes based on their morphological attributes,

with those exhibiting similar bulb shapes, sizes, and skin hues grouping more closely, indicating potential genetic or phenotypic similarities. The PCA score plot classified all genotypes into three distinct clusters, each influenced by specific traits. This indicates the presence of well-defined morphological variations that can be exploited for targeted breeding programs. Cluster III, the largest cluster, was positioned on the positive axis of PC1 and contained two sub-clusters of nine and 12 accessions each. With the exception of shape index, almost all the quantitative, and the qualitative traits, including shape of stem end (SSE, rounded), hue of color of dry skin (HCDS, reddish), degree of splitting into bullets (DSB, medium), position of maximum diameter of bulb (BPMD, middle), and bulb neck width (BWN, very narrow) exhibited significant positive load on PC1. Therefore, the majority of the accessions (cluster) aligning with the positive axis of PC1 indicates that they possess desirable levels of traits that contribute significantly to the variation captured by PC1. These findings illustrate the complex interplay of diverse traits in explaining the phenotypic variation within the onion germplasm and highlight the diversity available for potential breeding and genetic improvement programs. For each cluster, accessions with high mean values for a trait suggest their potential for hybridization to obtain a better segregate (*Singh et al., 2020*).

The dendrogram further revealed the distinct clusters of all the 29 accessions, constituting three groups, with Cluster III containing the highest number of accessions. The accessions (accessions 18 and 23) in Cluster I exhibited weak bulb splitting property, a trait desirable for commercial onion production as it enhances bulb integrity and marketability. They also exhibited uniformity in shape-related parameters, such as sphericity, aspect ratio, and shape index, suggests potential for mechanical harvesting and efficient post-harvest handling. Their pinkish dry skin color hue aligns with consumer preferences in certain markets, and thus these accessions hold a potential for high demand and value. Additionally, their small-sized bulbs with rounded root ends makes them ideal for specific culinary applications, such as markets favoring compact onions for ease of processing. These traits are valuable for crop breeding programs that may focus on developing onion varieties for both improved agronomic performance and meeting specific market demands. Strong bulb-splitting property was observed among accessions in Cluster II. This trait is important for breeding programs that may target developing cultivars suitable for propagation. This cluster was also characterized by a reddish dry skin color, which may be associated with higher antioxidant content, thus making the associated accessions suitable candidates for metabolomics studies to identify bioactive compounds that may be beneficial for human health. Their larger equatorial and polar diameters, along with maximum diameter positioned towards the stem end, indicate potential suitability for culinary applications requiring larger onion slices or bulk processing. These size and shape attributes could influence consumer preferences and market demand, particularly in regions where larger bulbs are favored. The medium splitting property observed in Cluster IIIA accessions makes them adaptable for both fresh market sales and production of propagation materials, and are flexible for breeding programs targeting developing cultivars with balance yield and storability. Their diverse qualitative traits, including variations in dry skin hue and base color, suggest genetic variability that could be explored in metabolomics studies

to identify pigment-associated bioactive compounds with potential health benefits. The predominance of larger bulb dimensions, such as equatorial and polar diameters exceeding 8 cm and 10 cm, respectively, aligns with consumer preferences in markets where bigger onions are favored for processing and bulk sales. Their bulb sphericity and shape indexes were near 1.0, indicating a uniform, and rounded shape, which contributes to enhancing mechanical harvesting efficiency and improves post-harvest handling. The diversity of qualitative characteristics observed in Cluster IIIB accessions, particularly in bulb splitting, dry skin coloration, and bulb size, suggest a significant level of genetic variability that could be explored for breeding. The dominance of brown dry skin coloration may indicate the presence of valuable bioactive compounds, making these accessions suitable for metabolomics studies focusing on antioxidant and phytochemical profiling. The medium to narrow neck width observed in most accessions in this cluster is beneficial for post-harvest curing and storage, as these traits are known to be associated with reduced moisture loss and increased shelf life. The uniformity in shape-related traits, such as bulb sphericity and shape index near 1.0, are suitable for mechanical handling, and thus desirable for large-scale onion production and automated processing. The observed variation in the position of the maximum diameter, ranging from the middle to the stem end, can influence slicing efficiency and consumer preference, in food processing industries. The larger polar and equatorial diameters (>8 cm and >6 cm, respectively) in most accessions in this cluster suggest the suitability of these accessions for markets favoring medium to large bulbs.

Correlation is a powerful statistical tool for understanding the relationships between traits (*Islam et al., 2019*). It provides insights into the interdependence and potential influence of traits on phenotypic variability. In this study, polar diameter exhibited a strong positive association with equatorial; diameter, bulb thickness, geometric diameter, arithmetic diameter, surface area, frontal surface area, and cross-sectional area, suggesting its critical role in determining overall bulb morphology. In line with the current results, previous study revealed a significantly positive correlation between bulb polar diameter, and bulb equatorial diameter (*Luitel et al., 2023*). However, in this study, the association between polar diameter and bulb shape index were not significant, which may be due to differences in both genotype and environment. Equatorial diameter showed a high positive correlation with frontal surface area, emphasizing its influence on bulb shape and visual appeal. Bulb thickness also displayed strong significant positive relationships with geometric diameter, arithmetic diameter, surface area, and cross-sectional area, reinforcing its importance in bulb size and volume. Correlated traits have genetic basis, with strong positive associations suggesting inherent strong association between the traits (*Visalakshi, Porpavai & Pandiyan, 2018*). Consequently, in crop breeding, selecting for one trait may tend to indirectly influence the other. Conversely, a strong negative association was observed between aspect ratio and bulb shape index, highlighting their inverse relationship in determining bulb form. Thus, in breeding, selecting for increased value for aspect ratio will likely cause a reduction in bulb shape index. However, aspect ratio showed no or weak correlation with other measured traits, such as frontal surface area, neck thickness, and various diameter measures, indicating its limited influence beyond shape-specific attributes. Shape index similarly showed no association with frontal surface area, suggesting these

traits may vary independently. Therefore, in breeding programs, these traits can be selected or targeted individually, with no potential link response, and allowing less likelihood of unintended effects. Overall, geometric mean diameter, arithmetic mean diameter, surface area, frontal surface area, and cross-sectional area exhibited high and significant positive correlations among themselves, demonstrating their potential genetic interconnectedness in defining bulb size and structure. This suggests that the accessions used in this study could be potential candidates for selection or further improvement in breeding programs targeting those specific traits.

## Variance components and heritability

In crop breeding, the knowledge about variance components, genetic gain, heritability is useful in facilitating genotype selection (*Amir et al., 2023*; *Patel et al., 2021*). The genetic variances estimated for the various bulb parameters suggest significant variability within the 29 onion genotypes. The GCV estimated provides a true suggestion about the magnitude of genetic variation. In this study, GCV and PCV for most parameters were high (>20%), suggesting the presence of appreciable amount of genotypic and phenotypic variability in the studied genetic materials. The higher values of PCV than the corresponding GCV were found for all traits, which suggests environmental conditions influence the expression of the traits studied (*Patel et al., 2021*). Narrow difference between PCV and GCV was observed for aspect ratio and bulb sphericity. Generally, values estimated for GCV/PCV ratio were found to be close to unity for all traits, except equatorial diameter, indicating that the expression of most of the traits are pre dominantly under genetic control (*Kumar & Kumar, 2017*). The findings in this study reveal that simple phenotypic selection of genotypes could be made for breeding. Besides, effective selection could be possible for most traits at an early stage in crop improvement programs. Understanding the magnitude heritability of trait is crucial in crop breeding to assess if traits that are of interest to breeders are heritable (*Akhter et al., 2021*; *Roka et al., 2024*). High to moderate broad sense heritability and genetic gain were estimated for several traits, such as thickness of neck, shape index, and aspect ratio, suggested that additive gene action underlie the expression of these traits (*Zewdu et al., 2024*); thus simple selection methods can be used. For traits such as polar diameter, transverse diameter, bulb sphericity which showed high to moderate heritability, but low genetic gain, implies that their improvement could be accomplished through hybridization or heterotic breeding. In a previous study (*Chavan & Jayappa, 2019*), high to moderate heritability and genetic advance (GA) estimates for thickness of neck, bulb diameter (cm), bulb length (cm), and bulb shape index were reported 49.40%, 67.10%, 50.20%, and 50.10%, with corresponding GA estimates of 23.02%, 30.45%, 21.29%, and 29.25, differing from values obtained in the present study. This differences may be as a result of genotype differences, and variation in environment or growing conditions. Several other studies have shown the potential of onion bulb traits as amenable to simple selection in improvement (*G, RP & Patil, 2021*; *Pujar et al., 2019*; *Singh et al., 2022*). The present study provides useful information requisite for selecting desirable traits and cross combinations in onion breeding.

## CONCLUSIONS

The primary goal of this study was to investigate the phenotypic variability of morpho-geometrical properties of onion bulbs among selected accessions. A range of phenotypic variation was found for the various traits studied. Based on principal component analysis, the 29 accessions were classified into three distinct clusters, with most of the studied characteristics varying widely among the accessions. These findings provide a potential for selection and breeding for improved traits. The results also revealed the potential of both the quantitative and qualitative characteristics in differentiating different onion accessions, and thus provides valuable insights for breeding programs and germplasm conservation. The observed phenotypic variability could be explored to develop onion varieties with desirable traits such as enhanced yield, and improved storage properties, which strongly align with industry needs. The current findings also provide a basis for understanding and selecting superior parental genotypes for breeding of high-performing onion cultivars that are tailored to diverse growing conditions and market demands. Accessions with desirable traits such as weak bulb splitting, uniform rounded shape, narrow neck width, and preferred skin coloration like brown or reddish may be prioritized for hybridization to enhance marketability and storability. Also, accessions with strong bulb splitting and larger bulb sizes may be considered for breeding programs aimed at developing varieties suitable for specific culinary and processing applications. Future studies could be designed to explore and identify candidate genomic regions or genes underlying natural variation of the studied traits in onion. Approaches such as genome-wide association studies (GWAS), bulk segregant analysis (BSA-seq.), and RNA sequencing (RNA-seq) could be used to identify candidate genes responsible for the natural variation observed in the studied traits of onion. These methods would enable the identification of genetic markers linked to key phenotypic traits, and facilitate marker-assisted selection (MAS) for more efficient and precise breeding programs. By exploring high-throughput sequencing and transcriptomic analysis, researchers could uncover the molecular mechanisms regulating bioactive compound accumulation, bulb development, and stress responses in onion germplasm. Such insights would accelerate the breeding of improved onion varieties with enhanced nutritional profiles, disease resistance, and environmental adaptability. Additionally, the influence of variability of quantitative and qualitative traits on metabolite composition in onion bulbs could be explored to facilitate selection of specific accessions based on known traits to enhance the nutritional and therapeutic potential of onion bulbs. For instance, knowledge about the correlation between bulb size, shape, and skin color with metabolite concentrations, such as flavonoids, polyphenols, and sulfur compounds could enable breeders to develop high-value onion varieties with superior health benefits. The significant association of traits like narrow neck width and reddish dry skin with key PCA components suggests that these morphological features may be linked to specific metabolic profiles, which could be validated through biochemical and molecular studies. This knowledge can inform breeding strategies aimed at enhancing both agronomic and functional properties, and ensuring that selected accessions not only meet market standards, but also contribute to dietary and medicinal applications. By integrating morphological

trait selection with metabolomic profiling, breeders can develop nutritionally optimized onion varieties tailored for both consumer preference and industrial applications. In terms of crop breeding, the high to moderate broad sense heritability and genetic gain estimated for several traits, including thickness of neck, shape index, and aspect ratio suggest that additive gene action underlie the expression of these traits; thus simple selection methods can be used.

### Funding

This research was carried out with the support of the "Research Program for Agricultural Science and Technology Development (Project NO. PJ01745401)", National Institute of Agricultural Sciences, Rural Development Administration, Republic of Korea. The funders had no role in study design, data collection and analysis, decision to publish, or preparation of the manuscript.

### Grant Disclosures

The following grant information was disclosed by the authors:
Research Program for Agricultural Science and Technology Development: PJ01745401.
National Institute of Agricultural Sciences, Rural Development Administration, Republic of Korea.

### Competing Interests

The authors declare there are no competing interests.

### Author Contributions

- Seong-Hoon Kim conceived and designed the experiments, performed the experiments, analyzed the data, prepared figures and/or tables, authored or reviewed drafts of the article, supervision, funding acquisition, and approved the final draft.
- Kanivalan Iwar conceived and designed the experiments, performed the experiments, analyzed the data, prepared figures and/or tables, authored or reviewed drafts of the article, and approved the final draft.
- JiWon Han conceived and designed the experiments, performed the experiments, analyzed the data, prepared figures and/or tables, authored or reviewed drafts of the article, provision of germplasm, and approved the final draft.
- Inchan Choi performed the experiments, analyzed the data, prepared figures and/or tables, authored or reviewed drafts of the article, and approved the final draft.
- Jaesu Lee performed the experiments, analyzed the data, prepared figures and/or tables, authored or reviewed drafts of the article, and approved the final draft.
- Kingsley Ochar conceived and designed the experiments, performed the experiments, analyzed the data, prepared figures and/or tables, authored or reviewed drafts of the article, and approved the final draft.

## Data Availability

The raw data is available in the Supplementary File.

## Supplemental Information

Supplemental information for this article can be found online at http://dx.doi.org/10.7717/peerj.19583#supplemental-information.

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
