# Peer review of "Variability of bulb morpho-geometrical properties in onion (*Allium cepa* L.) germplasm collections, using digital imaging tools"

_PeerJ, doi:10.7717/peerj.19583_

## Round 0.1 · original submission · Major Revisions

Dear Authors

The manuscript cannot be accepted for publication in its current form. It needs a major revision before publication. The authors are invited to revise the paper considering all the suggestions made by the reviewers. Please note that the requested changes are required for publication.

With Thanks

Reviewer 1 ·

Basic reporting

The manuscript needs major changes

Experimental design

The study demonstrates an acceptable experimental design.

Validity of the findings

The findings provide valuable information on the variability within the studied germplasm.

Additional comments

This paper presents a solid, if somewhat straightforward, study on the phenotypic characterization of onion germplasm. The use of digital imaging and ImageJ for measurement is a definite improvement over manual methods, offering efficiency and potential for more detailed analysis. The study's aim is clear, the methodology is sound (though more detail on the specific ImageJ tools used would be beneficial), and the results, while not groundbreaking, provide valuable information on the variability within the studied germplasm. The conclusion appropriately highlights the potential for breeding and further research. The text should be carefully proofread for grammatical errors and typos.
-Comments and Suggestions for Authors
Abstract
- Clarify "Geometrical Properties": The authors mention using linear dimensions to estimate other geometrical properties.
- It would be more impactful to give one or two specific examples of these associations. For example, "...strong significant positive associations were observed between…..and……
Introduction
- This introduction is quite comprehensive and well-structured.
- There's some repetition, particularly regarding the importance of onions and the impact of bulb morphology. For example, the economic importance is mentioned multiple times. Streamlining these sections would make the introduction more concise.
- Some sentences are a bit long and complex, making them slightly difficult to read. Breaking them down into shorter, more direct sentences would improve clarity.
- While many citations are good, some sentences feel overloaded. For example, the sentence about leading onion producers has a lot of citations.
Materials and methods
- Fertilizer Application: "20 mg ha-1" for cattle manure compost seems unusually low. It's more likely to be 20 tons per hectare. This needs to be checked and corrected. Also, while the fertilizer rates are given, it's not clear if these are the actual amounts applied or simply the recommended amounts. The text suggests the recommended amounts were followed, but this should be explicitly stated.
- Pest Control: While the pesticides are named, the application rates are missing. This is crucial information for reproducibility.
- Number of Bulbs Measured: The text states "freshly harvested bulbs, selected from each accession were used..." but doesn't specify how many bulbs per accession were measured. This is a critical detail for the statistical analysis and needs to be included.
- Statistical Methods: While the software is named, the specific statistical tests used (e.g., ANOVA, t-tests, etc.) are not mentioned. "Descriptive statistics" is too general. What specific descriptive statistics were calculated (mean, standard deviation, etc.)? What were the criteria for significance?
Results
- Some sentences could be more precise. For example, "tendency and degree of bulbs to split into bulblets ranged from week to strong" could be rephrased as "Bulb splitting tendency ranged from weak to strong.
- The use of accession numbers (e.g., 23P127, 23C68) is good, but it would be helpful to provide a table or supplementary information listing all the accessions and their origins or other relevant information. This would add context and value to the study.
- The study analyzes a large number of traits. It would be helpful to briefly explain the rationale behind selecting these specific traits. Why were these traits considered important for characterizing onion bulb diversity?
Discussion
- The authors attempt to explain the observed variability in both qualitative and quantitative traits, relating them to factors like marketability, consumer preference, and breeding potential.
- The discussion effectively links the findings to previous research, demonstrating a good understanding of the field. Citations are used appropriately.
- Some sentences are too long and complex, making them difficult to follow. Breaking them down into shorter, more concise sentences would enhance clarity.
- Some interpretations are too general. For example, "These results reveal the significant phenotypic diversity within the onion germplasm, and provide valuable insights for breeding programs..." What specific insights? How are they valuable? Provide concrete examples.
- The correlation analysis is mentioned, but the discussion is limited to a few specific correlations. A more comprehensive discussion of the correlation matrix, including both positive and negative correlations, would be valuable. Consider discussing the implications of these correlations for breeding.

Reviewer 2 ·

Basic reporting

Thank you for considering me to review the manuscript titled "Variability of bulb morpho-geometrical properties in onion (Allium cepa L.) germplasm collections". This manuscript uses digital imaging techniques to explore the phenotypic variability of onion (Allium cepa L.) germplasm collections, focusing on bulb morpho-geometrical properties. The study highlights the significance of bulb morphology in influencing consumer preferences, industrial processing, and mechanized post-harvest handling. Unlike previous research that relied on manual measurement methods, this study employs high-throughput phenotyping tools, such as digital cameras and ImageJ software, for rapid and accurate assessment of bulb traits. A total of 29 onion accessions were evaluated based on qualitative and quantitative characteristics, revealing significant variability in bulb dimensions and shape. Principal component analysis grouped the accessions into distinct clusters, and strong trait correlations were identified, emphasizing the genetic diversity within the germplasm. These findings provide valuable insights for onion breeding programs, aiding in the development of cultivars with improved agronomic performance and market suitability. Overall, this study underscores the potential of digital imaging technologies in modern phenotypic characterization, contributing to more efficient crop improvement strategies.

Specific comments:
Title
The title is clear but could benefit from specifying the focus on the use of digital imaging tools. "Phenotypic variability of bulb morpho-geometrical properties in onion (Allium cepa l.) germplasm collections using digital imaging tools"

Abstract
The abstract is well written but could be improved by adding a brief mention of the imaging setup and how the data were processed. Also, the results could be more concise by avoiding repetition. A stronger conclusion could be added to directly tie the results to practical applications, such as the breeding potential for improved onion cultivars or industrial use.

Introduction
The introduction provides a comprehensive background on onion importance as a crop and the significance of bulb morphology. The first paragraph could be streamlined. Avoid repeating general facts such as onion global production that don't directly connect to the study objective.
A clearer justification for why studying the phenotypic variability of bulb properties is needed would strengthen the introduction. What gaps in the current knowledge does this study address?

Materials and Methods
The methodology is well-detailed. Just the use of digital cameras and ImageJ is described well, but the setup for capturing the images (such as camera settings, angle, and distance from the bulb) should be detailed more precisely for reproducibility.

Results
The variability in qualitative traits is well presented. However, the data could be summarized more concisely. There is some redundancy when reporting ranges for each trait. For instance, the ranges for polar and equatorial diameters are mentioned twice with similar wording.
The PCA section provides good insights, but the explanation of results could be clearer. The use of loading plots and the connection between quantitative and qualitative traits should be more explicitly linked to the overall findings.
The resolution of figures, especially Figure 2-5, must be improved

Discussion
While the section describes various qualitative traits, it could be more explicit about how these findings can be used to guide breeding programs. For instance, explain in more detail how specific qualitative traits can be selected for breeding to improve specific traits like market appeal or storage life. Also, which accessions with particular traits should be prioritized for hybridization? What are the next steps for genetic studies that could aid these breeding efforts?
Briefly mention how similar studies or breeding programs have used these findings in the past, either for onions or other crops. This would help solidify the practical applications of the study.
The discussion of traits should tie back more directly to how the observed phenotypic variability could lead to improved onion cultivars. This could be further expanded by suggesting potential crosses or specific traits to focus on in future breeding.
The section includes some comparisons with previous studies, but these could be linked more tightly to the current results. The comparison of ranges of polar and equatorial diameters with other studies is useful, but the impact of these differences on breeding or market selection should be emphasized more clearly.
While the results of PCA are presented, the discussion of these results could be expanded to include more detailed interpretations of the PCA loading plots and score plots.
The discussion of clusters could benefit from a more detailed breakdown of what traits distinguish the different clusters. The current description provides some information about the clustering but could be more explicit about which traits are associated with each cluster and what that means for breeding programs.

The suggestion of exploring molecular markers is an excellent idea but could be developed further. You could discuss the potential for specific markers associated with key traits and how this could speed up the breeding process.
The suggestion to explore the relationship between morphological traits and biochemical composition could be expanded.

Conclusion
The conclusion could benefit from more emphasis on the broader applications of the study findings. Specifically, it could better highlight the practical implications of the phenotypic variability observed and how it will influence both breeding practices and industry needs in the near future.
Mention the potential for molecular research more explicitly. For example, you could state that future studies could integrate genomic tools to identify the genetic basis of key traits and aid in marker-assisted selection for efficient breeding programs. It would be helpful to outline some potential methods for these genomic studies, such as the use of genome-wide association studies (GWAS) or RNA sequencing to identify candidate genes for key traits. Additionally, how might these methods contribute to the speed and precision of breeding programs?
The suggestion to explore the impact of phenotypic traits on metabolite composition is a promising avenue for future research. It would strengthen the conclusion if you tied this more closely to the findings of your study, such as investigating how size, shape, or color affect the levels of bioactive compounds in the bulbs.

Reference
Please ensure that the formatting is uniform across all references. Some references are not consistently formatted; see for example, lines 526 and 536. Also, the journal name should be revised throughout the list; see lines 567, 571, 582, and 603. The titles of papers in the reference list should be consistently formatted. Some references capitalize the first letter of each word (548, 552,….), while others do not.
Lines 510, 524, 531, 534, 553, 560, 564, 576, 592, 607, 613, 617, 621, …….: the scientific names should be italicized and also throughout the manuscript
512: The genus name should always begin with a capital letter, also throughout the manuscript.

Experimental design

Experimental design is appropriate

Validity of the findings

The findings and interpretations need improvement

Reviewer 3 ·

Basic reporting

The research article focuses on the diversity of morpho-geometrical characteristics of bulbs in onion (Allium cepa L.) germplasm collections.

The article consistently employed a clear and professional tone in its use of the English language.
The literature references utilized were pertinent, and the citations were conducted with appropriate context.

The structure of the article was suitable according to the guidelines, and the figures and tables utilized were appropriate; however, some minor corrections are necessary.

Experimental design

he primary research conducted elucidates the significance of phenotypic characterization of onion germplasm. The article emphasizes a study designed to explore the phenotypic variability among 29 onion accessions, focusing on ten qualitative and twelve quantitative bulb characteristics.

In this context, the research sought to identify an effective digital solution for scanning these 29 onion accessions. Previously, manual tools such as Vernier calipers were employed to measure onion bulb parameters, a process that proved to be time-consuming. The advent and utilization of phenomics tools, such as digital cameras, offer a more efficient means for the rapid phenotypic characterization of these bulb parameters.

The research questions were clearly articulated and significant. The investigations were conducted rigorously, maintaining high ethical standards throughout the process.

Methodological information was provided in detail and was credible.

Validity of the findings

The article emphasizes the utilization of digital cameras as a more effective method for the swift phenotypic characterization of bulb parameters. The data presented in the study is both reliable and substantial. The conclusion is clearly articulated, and the supporting results are adequately provided.

Corrections are required as follows:
1. The English sentence in line 382 requires revision.
2. In lines 437 and 438, please provide a complete description of the abbreviations utilized.
3. The reference in line 522 should be cited in accordance with the guidelines.
4. Figures 2, 3, 4, 5, and 6 appear to be unclear.

---

## Round 0.2 · Major Revisions

Dear Authors,

Given that this study aims to characterize the phenotypic variability among onion accessions, an ANOVA for each trait should be performed to assess if there is a significant difference among the accessions. The ANOVA could also calculate the percentage of variance explained by accession to accession differences (aka broad sense heritability).

With Thanks

Reviewer 1 ·

Basic reporting

no comment

Experimental design

no comment

Validity of the findings

no comment

Additional comments

The authors have made the changes I suggested in the last review. I recommend its publication in this journal.

Reviewer 2 ·

Basic reporting

The authors have addressed all of the previous comments and significantly improved the manuscript

Experimental design

The experimental design is appropriate

Validity of the findings

The findings are supported by robust data

Reviewer 3 ·

Basic reporting

Revisions have been made

Experimental design

Revisions have been made

Validity of the findings

No comments as the corrections are done

Additional comments

NA

---

## Round 0.3 · accepted · Accept

Dear Authors,

I am pleased to inform you that the manuscript has improved after the last revision and can be accepted for publication.

Congratulations on accepting your manuscript, and thank you for your interest in submitting your work to PeerJ.

With Thanks